# Identification of the Time Period during Which BMP Signaling Regulates Proliferation of Neural Progenitor Cells in Zebrafish

**DOI:** 10.3390/ijms24021733

**Published:** 2023-01-15

**Authors:** Hung-Yu Shih, Chia-Wei Chang, Yi-Chieh Chen, Yi-Chuan Cheng

**Affiliations:** 1Graduate Institute of Biomedical Sciences, College of Medicine, Chang Gung University, Taoyuan 33302, Taiwan; 2Department of Biological Science, College of Science, Engineering & Technology, Utah Tech University, St. George, UT 84770, USA; 3Neuroscience Research Center, Chang Gung Memorial Hospital at Linkou Medical Center, Taoyuan 333423, Taiwan; 4Department of Neurology, Chang Gung Memorial Hospital at Linkou Medical Center, Taoyuan 333423, Taiwan

**Keywords:** BMP, Notch, neural progenitors, zebrafish

## Abstract

Bone morphogenetic protein (BMP) signaling regulates neural induction, neuronal specification, and neuronal differentiation. However, the role of BMP signaling in neural progenitors remains unclear. This is because interruption of BMP signaling before or during neural induction causes severe effects on subsequent neural developmental processes. To examine the role of BMP signaling in the development of neural progenitors in zebrafish, we bypassed the effect of BMP signaling on neural induction and suppressed BMP signaling at different time points during gastrulation using a temporally controlled transgenic line carrying a dominant-negative form of Bmp receptor type 1aa and a chemical inhibitor of BMP signaling, DMH1. Inhibiting BMP signaling from 8 hpf could bypass BMP regulation on neural induction, induce the number of proliferating neural progenitors, and reduce the number of neuronal precursors. Inhibiting BMP signaling upregulates the expression of the Notch downstream gene hairy/E(spl)-related 2 (*her2*). Inhibiting Notch signaling or knocking down the Her2 function reduced neural progenitor proliferation, whereas inactivating BMP signaling in Notch-Her2 deficient background restored the number of proliferating neural progenitors. These results reveal the time window for the proliferation of neural progenitors during zebrafish development and a fine balance between BMP and Notch signaling in regulating the proliferation of neural progenitor cells.

## 1. Introduction

Neural progenitor cells are proliferative and multipotent cells that can differentiate into various cell types, such as neurons, oligodendrocytes, and astrocytes of the central nervous system [1]. These cells are generated from the neuroectoderm during early embryonic development. Studies in *Xenopus* and zebrafish demonstrated that during early gastrulation, secreted neural inducers, such as Noggin, Follistatin, and Chordin, inhibit bone morphogenetic protein (BMP) signaling at the dorsal side of the animal cap, where they repress the epidermal fate and induce the neural fate of the ectoderm. In contrast, the ventral ectoderm expresses a high level of BMP and differentiates into the epidermis. This process, termed neural induction, specifies the neuroectoderm within the ectoderm and is conserved in vertebrates. However, other signals, such as fibroblast growth factor (FGF) and Wnt, have also been demonstrated as key modulators in neural induction (reviewed in [2]). These signaling pathways and their downstream transcription factors not only govern neural induction, but also regulate the competence and specification of neural progenitor cells in the neuroectoderm, as well as the formation and expansion of neural plates (reviewed in [3,4]). Therefore, spatial and temporal regulation studies, such as transplantation, tissue-specific promoter-driven gene expression, and electroporation, have been performed in animal models to clarify the role of these signaling pathways in neural induction and the development of neural progenitor cells.

BMPs belong to the transforming growth factor-beta (TGF-β) superfamily, which mediates a diverse array of cellular processes, including proliferation, differentiation, and apoptosis [5]. BMPs bind to type 1 and type 2 serine/threonine transmembrane receptors (BMPR1 and BMPR2, respectively) in recipient cells. Upon binding, BMP receptors phosphorylate carboxy-terminally located serine residues in downstream transcription factors SMAD1/5/9. Upon phosphorylation, these transcription factors form complexes with the common mediator SMAD protein SMAD4, and the complex subsequently accumulates in the nucleus to activate the transcription of downstream genes, such as inhibitors of differentiation (*Id*) and *Msx* [6,7]. BMPs also initiate SMAD-independent signaling pathways, such as the MAPK, PI3K-AKT, protein kinase C, and Rho-GTPase signaling pathways (reviewed in [8]). BMP signaling exhibits pleiotropic effects on the developing nervous system and adult homeostasis. The BMP signaling gradient establishes the dorsoventral axis and negatively regulates neural induction to suppress neuroectoderm formation ([9,10], reviewed in [11]). Once the neural tube closes, BMP functions as a morphogen to regulate neural tube patterning and specify the neuronal subtypes in the dorsal neural tube (reviewed in [12]). Furthermore, BMP signaling regulates neurogenesis and gliogenesis. BMP2 and BMP4 have been shown to promote astrocyte formation by suppressing oligodendrocyte formation. [13,14]. BMP2 restricts proliferation and induces neuronal differentiation of neuroepithelial cells [15], whereas BMP4 overexpression stimulates neuronal differentiation [16]. In contrast, BMP2, 4, or 7 suppress the generation of olfactory receptor neurons [17] and reduce neurogenesis in cultured mouse fetal telencephalic neuroepithelial cells [18,19]. However, the role of BMP signaling in regulating the development of neural progenitor cells has mostly been demonstrated in in vitro studies and requires further confirmation in vivo.

In this study, we bypassed the effect of BMP signaling on the regulation of embryonic development before early gastrulation, including dorsoventral patterning, mesoderm formation, and neural induction, and examined the specific role of BMP signaling in the development of neural progenitors. We inhibited BMP signaling at distinct time points during zebrafish gastrulation using a transgenic fish line that carries the dominant-negative form of the Bmp type 1 receptor (DN-BmpR1aa) and confirmed the results using a chemical inhibitor of BMP signaling, dorsomorphin homologue 1 (DMH1). We defined the time point for the effect of BMP signaling during neural progenitor development and discovered that Notch-Her2 signaling underlies this regulation. 

## 2. Results

### 2.1. Bmpr1aa Is Expressed in the Developing Nervous system during Gastrulation and Early Segmentation

In zebrafish, neural induction occurs during gastrulation (from 5.25 h post-fertilization (hpf) to 10 hpf). During this period, the neural progenitors are specified from the ectoderm [20] and then at the end of gastrulation (10 hpf), the neuronal precursor cells are formed within neural progenitors by activating the expression of proneural genes [21]. Neural progenitors give rise to neuronal precursors that generate a variety of neurons in the nervous system during segmentation. We first analyzed the expression patterns of one of the Bmp type 1 receptors, *bmpr1aa*, during the gastrulation and early segmentation stages. The results of whole-mount in situ hybridization demonstrated that *bmpr1aa* was ubiquitously expressed in the entire embryo at 5.3 hpf and 6 hpf, and from 8 hpf onward, was expressed in the presumptive neuroectoderm and developing mesoderm (Figure 1A). At the end of gastrulation (10 hpf), strong *bmpr1aa* expression was detected in the developing forebrain, midbrain, trigeminal primordium, and the sensory and motor neurons of the spinal cord (Figure 1B). During the early segmentation stage (11 hpf), *bmpr1aa* was strongly expressed in the developing nervous system (Figure 1). This result suggested that *bmpr1aa* participates in the formation of neural progenitor cells after neural induction.

### 2.2. Temporally Inhibiting BMP Signaling by Dominant-Negative Bmpr1aa and DMH1

BMP signaling negatively regulates neural induction [22], and the disruption of BMP signaling at this stage disrupts neural developmental progress, thus hindering the analysis of BMP signaling in neural progenitor cells. In order to temporally inhibit BMP signaling, we adapted the inducible transgenic zebrafish line *Tg *(*hsp70l: dn-bmpr1aa-eGFP*)** [23] which carries a dominant-negative form of Bmpr1aa (DN-Bmpr1aa) driven by a heat-shock promoter, thus facilitating temporal inactivation of BMP signaling. We first evaluated the effectiveness of this transgenic line in alerting BMP signaling by analyzing the expression of *eve1*, which is a downstream target of BMP signaling and is decreased in *bmp* mutants [24,25]. Heat-shock activation of DN-Bmpr1aa at 5.3 hpf inhibited BMP signaling, as observed by the downregulation of *eve1* expression at 6 hpf and diminished *eve1* expression at 7 hpf (Figure 2A,C). These results demonstrated that overexpressing DN-Bmpr1aa inhibits BMP-Bmpr1aa signaling 0.7 h after induction of heat-shock.

The BMP type 1 receptors initiate both Smad-dependent and Smad-independent signaling cascades to control various cellular processes [8,26]. To further confirm the regulation of dominant-negative Bmpr1aa, we used the chemical inhibitor dorsomorphin homologue 1 (DMH1), which specifically blocks Smad-dependent signaling [27]. We tested different concentrations of DMH1 at different time points during embryonic development and discovered that treatment with 0.5 μM DMH1 sufficiently inhibited BMP-Smad signaling demonstrated by the reduced *eve1* expression (Figure 2B,D). Treatment with 0.5 μM DMH1 reduced *eve1* expression, which was identical to the effect of DN-Bmpr1aa overexpression at the same time point. Hereafter, we used 0.5 μM DMH1 in the following experiments. 

Id3 and Msx1b (previously named Msxb) are direct downstream targets of the BMP-Smad signaling pathway. Upon binding of BMPs, BMP receptors phosphorylate Smad1/5/9 and the phosphorylated Smad1/5/9 and Smad4 complex directly bind to the 5ʹ regulatory elements of *id3* and *msx1b* and induce their expression [28,29,30]. To further confirm that dominant-negative Bmpr1aa and DMH1 were sufficient to inhibit BMP-Smad signaling, we examined the effect of BMP inactivation on the expression of *id3* and *msx1b*. The result demonstrated that heat-shock activation of DN-Bmpr1aa or DMH1 treatment at 5.3 hpf inhibited the expression of *id3* and *msxb* at 6, 7, and 9 hpf, confirming that DN-Bmpr1aa and DMH1 were sufficient to inhibit BMP-Smad signaling (Appendix A).

### 2.3. Inhibiting BMP Signaling from 8 hpf Bypasses the Effects of BMP Signaling on Dorsoventral Patterning and Neural Induction

Next, we identified the time point at which BMP signaling was inhibited, without affecting neural induction. Inactivation of BMP signaling during neural induction results in a reduced surface ectoderm and an expansion of the neuroectoderm [10]. The transgenic line *Tg* (*hsp70l: dn-bmpr1aa-eGFP*) and DMH1 were used to inhibit BMP-BmpR1aa-Smad signaling. We performed heat-shock induction or treatment with DMH1 at 5.3, 8, or 9 hpf for one hour, and harvested the embryos at 10 hpf for examination. In situ hybridization analysis for the expression pattern of neuroectoderm markers *sox2* and *sox3* demonstrated that inactivating BMP-Bmpr1 signaling at 5.3 hpf caused the expansion of *sox2* and *sox3* expression at 10 hpf (Figure 3A,B, arrowheads). In contrast, blocking BMP-BmpR1 signaling at 8 or 9 hpf did not alter the expression patterns of *sox2* and *sox3* at 10 hpf (Figure 3A,B). This result demonstrates that inhibiting BMP signaling after 8 hpf did not affect neural induction.

Inhibiting BMP signaling before early gastrulation induces the expansion of the neuroectoderm and loss of the epidermal ectoderm ([24], reviewed in [31,32]). We performed heat-shock activation of DN-Bmpr1aa or DMH1 treatment at 4 hpf, examined the expression of the surface ectoderm marker *tp63* at 50% epiboly (5.3 hpf), and discovered that inhibiting BMP signaling at this time point was sufficient to inhibit the formation of surface ectoderm, as evidenced by reduced *tp63* expression (Appendix A). Next, we inhibited BMP signaling at 8 or 9 hpf and found that the expression of *tp63* was not altered at 10 hpf (Figure 4). Taken together, these data indicated that inhibiting BMP-Bmpr1aa-Smad signaling from 8 hpf bypassed the effect of BMP signaling on dorsoventral patterning and neural induction in zebrafish embryos.

### 2.4. BMP Signaling Is Essential for Suppressing Neural Progenitor Proliferation at 8 hpf to 10 hpf

Inactivating BMP signaling after 8 hpf did not affect the gross expression pattern of *sox2* and *sox3.* However, we noticed that the *sox2*-expressing areas were slightly increased (although not statistically significant) by BMP inactivation (Figure 3). This might be because inactivating BMP alters the development of neural progenitors at 8–10 hpf. At this stage, neural progenitors increase in number by proliferation. Accordingly, we inactivated BMP-BmpR1 signaling at 8–10 hpf and examined neural progenitor cell proliferation. In addition to measuring the *sox2*-expressing areas (Figure 3C), we conducted quantitative reverse transcriptase PCR (qRT-PCR) to investigate the altered *sox2*-expression level after BMP signaling inactivation. The results demonstrated that inactivation of BMP-Bmpr1aa-Smad signaling upregulated *sox2* expression significantly (Figure 5). In addition, we counterstained *sox2* with the proliferating marker, phospohryl histone 3 (pH3), and discovered that pH3-positive neural progenitors were increased in BMP-signaling-inactivated embryos (Figure 5). This result demonstrated that BMP signaling is essential for suppressing neural progenitor proliferation. We also examined the apoptosis of neural progenitors in BMP-signal-inactivated embryos by counterstaining *sox2* with the apoptotic marker cleaved caspase 3 and found no significant alteration in the number of apoptotic cells caused by inactivation of BMP signaling (Appendix A). These results demonstrate that inhibiting BMP signaling induces the proliferation of neural progenitors without affecting cell apoptosis.

### 2.5. BMP Signaling Is Essential for Neuronal Precursor Development between 8 hpf to 10 hpf

Neuronal precursor cells differentiate from neural progenitors from 10 hpf, as indicated by the onset of expression of many proneural genes, such as *neurog1*, during zebrafish development [21]. Accordingly, we examined the effect of inactivation of BMP-Bmpr1aa-Smad signaling on the formation of neural progenitors by examining *neurog1* expression. Inhibition of BMP-Bmpr1aa-Smad signaling at 8 to 10 hpf reduced the number of *neurog1*-positive cells (Figure 6), indicating that BMP signaling is essential for the formation of neuronal precursors. These results suggest that BMP-Bmpr1aa-Smad signaling is required for repressing neural progenitor proliferation and is essential for neuronal differentiation after neural induction.

### 2.6. Suppression of Notch Signaling Abolished the Phenotype Caused by BMP-Signaling Inactivation

BMP signaling interacts with other molecular signaling pathways to control diverse developmental events [33]. Notch signaling is a critical regulator of neural progenitor proliferation and the suppression of neuronal differentiation [34]. A previous study showed that BMP signaling plays a role opposite to Notch signaling in governing neuronal differentiation in the olfactory bulb of developing chick embryos [35]. To evaluate whether Notch signaling plays a role in BMP-signaling-mediated neural progenitor proliferation and neuronal differentiation, we inhibited BMP signaling and concomitantly inhibited Notch signaling activity using a chemical inhibitor, N-[N-(3,5-Difluorophenacetyl)-L-alanyl]-S-phenylglycine t-butyl ester (DAPT), from 8 hpf to 10 hpf. These embryos were analyzed for *sox2* expression using in situ hybridization and qRT-PCR. DAPT treatment abolished the upregulation of *sox2* expression caused by BMP signal inhibition (Figure 7). In addition, blocking Notch signaling reduced neural progenitor proliferation (Figure 7) and abolished the upregulation of proliferation of neural progenitors caused by inactivation of BMP-Bmpr1-Smad signaling (Figure 5). This result suggests an interaction between BMP and Notch signaling in regulating neural progenitor proliferation after neural induction.

### 2.7. BMP and Notch-Her2 Signaling Regulate the Proliferation of Neural Progenitor Cells

Our results demonstrated that inactivating BMP signaling promoted the proliferation of neural progenitors, whereas inactivation of Notch signaling reduced the proliferation of neural progenitors. Although inactivation of Notch signaling could abolish the effect of inactivation of BMP signaling on neural progenitors, the action of these two signaling pathways could be interactive or parallel. BMP signaling has been reported to antagonize Notch signaling by inhibiting the expression of downstream transcription factors to balance the proliferation and differentiation of neural progenitor cells [35,36,37]. To analyze the interaction between BMP and Notch signaling in the proliferation of neural progenitors, we examined the function of hairy/E(spl)-related 2 (Her2), which is a homologue of mammalian *Hes5* and is a downstream target of Notch signaling. It has been shown to promote neural progenitor proliferation and is essential for inhibiting neuronal differentiation [38,39] in BMP-signaling-inactivated embryos. We inactivated BMP-Bmpr1-Smad signaling at 8 to10 hpf by activating DN-BmpR1aa or treatment with DMH1, followed by qRT-PCR analysis for *her2* expression at 10 hpf. Expression of *her2* was upregulated by inactivation of BMP signaling (Figure 8A,D), indicating that BMP-BmpR1-Smad signaling is required for suppressing *her2* expression. Knocking down Her2 function by injecting *her2* morpholino at the one-cell stage reduced the proliferation of neural progenitors, as described by Cheng et al. (Figure 8B,F), [38], whereas inactivation of BMP signaling in Her2 knockdown embryos recovered the number of proliferating neural progenitors (Figure 8). Taken together, although inactivation of BMP signaling induced *her2* expression, in embryos lacking Her2 expression, inactivation of BMP signaling could still upregulate the proliferation of neural progenitors. This result indicated that BMP and Notch signaling regulate the proliferation of neural progenitors in both a parallel and an interactive manner, and a fine balance between BMP and Notch signaling pathways is essential for the proper proliferation of neural progenitors.

## 3. Discussion

Precisely controlled proliferation and differentiation of neural stem/progenitor cells ensures the proper number and type of neurons to be generated in the brain. Impaired development of neural stem/progenitor cells can cause defective brain size, neurological disorders, and brain tumors. Understanding the proliferation and differentiation of neural stem/progenitor cells could provide a potential therapeutic approach for stem-cell-based therapies. The coordination of many molecular signaling pathways with precise spatial and temporal control is essential for the development of neural stem/progenitor cells. However, the time window of a given molecular signaling pathway in regulating this event has not been fully defined. In the present study, we discovered a critical time window for BMP signaling in the regulation of neural progenitor proliferation in zebrafish. Inactivation of BMP-Bmpr1a-Smad signaling from mid-gastrulation (8 hpf) to end-gastrulation (10 hpf) promotes the proliferation of neural progenitor cells and inhibits the differentiation of neuronal precursors. To our knowledge, this is the first in vivo study to define the time window of BMP-Bmpr1a-Smad signaling in regulating embryonic neural progenitor cell proliferation.

We also discovered that BMP-Bmpr1a-Smad and Notch-Her2 signaling regulate the proliferation of neural progenitors in an opposite manner—inhibiting BMP-Bmpr1a-Smad signaling promotes neural progenitor proliferation, whereas inactivating Notch-Her2 signaling reduces neural progenitor proliferation. Although inactivating BMP signaling upregulates *her2* expression, suggesting an interaction between these two signaling pathways in regulating neural progenitor proliferation, inactivating BMP signaling upregulates neural progenitor proliferation in Her2 knockdown embryos, suggesting that these two signaling pathways may also regulate neural progenitor proliferation in a parallel manner. Taken together, a proper balance between BMP and Notch signaling is essential for the development of neural progenitors.

## 4. Materials and Methods

### 4.1. Ethics Statement

All experiments were performed in strict accordance with the standard guidelines for zebrafish work and were approved by the Institutional Animal Care and Use Committee of Chang Gung University (IACUC approval numbers: CGU07-74, CGU11-118, and CGU12-118).

### 4.2. Fish Maintenance and Transgenic Lines

AB (wild-type) zebrafish and transgenic line *Tg* (*hsp70l:dnBmpr1aa-GFP*) [23] were purchased from the Zebrafish International Resource Center (Eugene, OR, USA) and raised, maintained, and paired under standard conditions. Embryos were staged according to the hours post-fertilization [40].

### 4.3. Heat Shock and DMH1 Treatment

Batches of wild-type and transgenic embryos were strictly staged, as previously described [40] to ensure minimal developmental asynchrony. Wild-type and transgenic embryos were treated with DMH1 (Merck, Taipei, Taiwan) or induced by heat shock at 37 °C.

### 4.4. Morpholinos

*her2* morpholino (CTGGAAAGAGAAGGTAAAAGTTTGG) was purchased from Gene Tools LLC. (Gene Tools, LLC., Philomath, OR, USA). Morpholinos with a random sequence that does not bind to the zebrafish genome (CCTCTTACCTCAGTTACAATTTATA) and with 5 mismatched bases to the *her2* morpholino (CTaGAAAaAGAAGaTAAAAaTTTaG; mismatches are displayed by lowercase letters) were used as controls. Microinjection of morpholinos into zebrafish blastomeres was performed during the one- to two-cell stages. A detailed procedure of *her2* knockdown has been described by Cheng et al. [38].

### 4.5. Histological Analysis

Digoxigenin-UTP-labeled riboprobes were synthesized according to the manufacturer’s instructions (Roche, Taipei, Taiwan) and in situ hybridization was performed as described previously. Color reactions were conducted using NBT/BCIP substrate (Roche, Taipei, Taiwan). To minimize variation between the control and experimental groups, we used embryos produced by a single pair of parents and used the same number of embryos for the control and experimental groups. The groups were compared under the same experimental conditions at the same time, and color reactions were initiated and stopped at precisely the same time. For immunohistochemistry, zebrafish embryos were blocked in 5% goat serum and incubated with a rabbit phospho-histone H3 antibody (1:200, Merck Millipore, distributed by Level Biotechnology Inc., Taipei, Taiwan). Phosphorylated histone H3 was detected using red-fluorochrome-conjugated secondary antibodies (Invitrogen, distributed by Level Biotechnology Inc., Taipei, Taiwan). Embryos were mounted in the Vectashield mounting medium (Vector Laboratories, Inc., distributed by Asia Bioscience Co., LTD., Taipei, Taiwan). Phospho-histone H3-positive cells were manually counted. To quantify *sox2* expression by in situ hybridization, *sox2*-positive areas were selected and measured using ImageJ software. To detect apoptotic cells in the neuroectoderm, immunohistochemistry using rabbit monoclonal anti-active caspase-3 antibody (Abcam, distributed by Blossom Biotechnologies, Inc., Taipei, Taiwan) was applied to samples already stained with *sox2* riboprobe using in situ hybridization. The caspase-3 antibody was detected using goat anti-rabbit IgG HRP (Invitrogen, distributed by Level Biotechnology Inc., Taipei, Taiwan), and 3, 3′-diaminobenzidine was used as a substrate for secondary antibody-conjugated HRP (Amresco, distributed by Protech Technology Enterprise Co., Ltd., Taipei, Taiwan). Phospho-histone H3- and caspase-3-positive cells were manually counted.

### 4.6. RNA Isolation and cDNA Synthesis

RNA was isolated from 20 embryos in each experiment. Samples collected from each experiment were suspended in 500 μL TRIzol reagent (Invitrogen, distributed by Level Biotechnology Inc., Taipei, Taiwan) and homogenized using a 25-gauge needle, followed by the addition of 250 μL chloroform and centrifugation to separate the aqueous phase. The aqueous solution was mixed with a half-volume of acid phenol and chloroform and subjected to centrifugation for 15 min at 4 °C. The phenol-chloroform extraction was repeated twice. RNA was extracted using 250 μL isopropanol and 70% ethanol, and the mixture was centrifuged for 30 min at 4 °C. The ethanol was then removed, and the RNA pellet was resuspended in 12 μL DEPC H_2_O. The first strand of cDNA was reverse-transcribed using the SuperScript III First-Strand Synthesis System (Applied Biosystems, distributed by Creative LifeSciences, Taipei, Taiwan).

### 4.7. Quantitative Analysis

For quantitative reverse transcriptase PCR (qRT-PCR), embryos were homogenized in TRIzol reagent (Invitrogen, distributed by Level Biotechnology Inc., Taipei, Taiwan), and total RNA was extracted using a standard method. cDNA was synthesized from total RNA through oligo(dT) priming using RevertAid First Strand cDNA synthesis kit (Bionovas, distributed by Scientific Biotech Corp., Taipei, Taiwan). qPCR was performed on an ABI StepOneTM Real-Time PCR System (Applied Biosystems, distributed by Creative LifeSciences, Taipei, Taiwan) using SYBR Green (ABclonal Technology, distributed by Interlab Co., LTD., Taipei, Taiwan). Primers for *sox2* (F:5’-CACCAACTCCTCGGG AAACA-3′; R:5′-AATGGTCGCTTCTCGCTCTC-3′), *her2* (F:5′-GAGATGGCTGTTATTTACCT-3′; R:5′-TTGGGTTTGTTTGTGAGC-3′), and *GAPDH* (F:5′-ACCCGTGCTGCTTTCTTGAC-3′; R:5′-GACCAGTTTGCCGCCTTCT-3′) were used. Gene expression levels were normalized to *GAPDH* and assessed using comparative Ct (40 cycles), according to the manufacturer’s instructions (7500 Real-Time PCR System, Applied Biosystems, distributed by Creative LifeSciences, Taipei, Taiwan).

Statistical analysis was performed using Student’s *t*-test with Microsoft Excel 2016. The significance level was set at *p* < 0.05. All reactions were performed in triplicate for each sample.

## Figures and Tables

**Figure 1 ijms-24-01733-f001:**
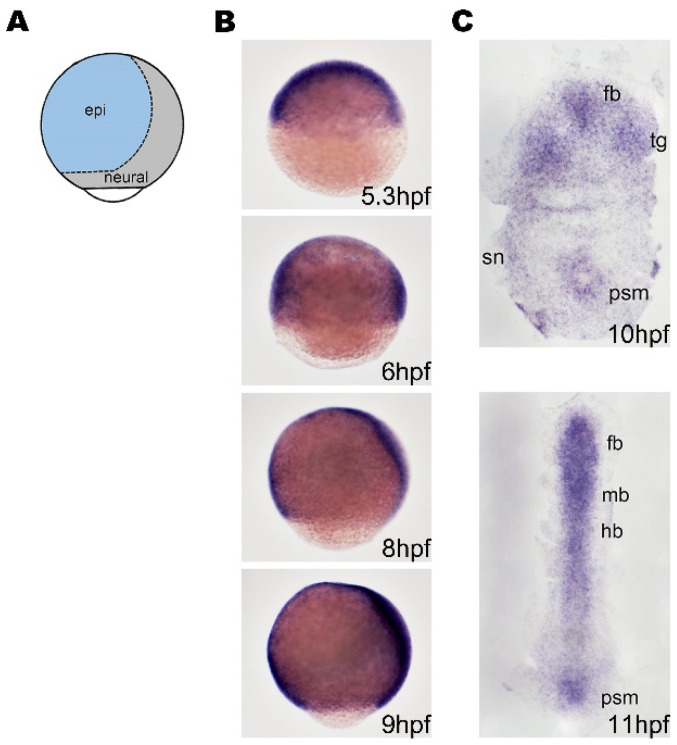
The expression of *bmpr1aa* during gastrulation and early segmentation. (**A**) A schematic illustration of the lateral view of zebrafish embryo showing the regions of neuroectoderm (neural) and surface ectoderm (epi) with dorsal to the right. (**B**) Zebrafish embryos were analyzed using whole-mount in situ hybridization. These images show the lateral view of the embryos with dorsal to the right and ventral to the left. (**C**) Dorsal view of flat-mounted embryos with anterial to the top. (**B**) The *bmpr1aa* is ubiquitously expressed before 8 hpf, and gradually expressed in neural tissue from 8 hpf. (**C**) *bmpr1aa* is expressed in developing brain at 10 hpf (end of gastrulation) and 11 hpf (early segmentation). fb, forebrain; mb, midbrain; hb, hindbrain; epi, epidermal; sn, sensory neuron; psm, presomatic mesoderm.

**Figure 2 ijms-24-01733-f002:**
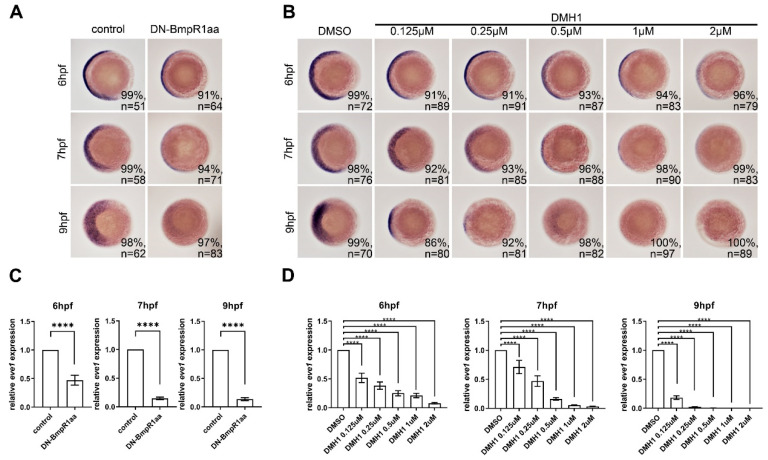
DN-Bmpr1aa and DMH1 sufficiently inhibited expression of *eve1* at different time points. All panels are animal pole views with ventral to the left. Expression of *eve1* was examined using in situ hybridization. (**A**) *Tg* (*hsp70l:dnBmpr1aa-GFP*) embryos were heat shock-treated at 5.3 hpf and harvested at 6, 7, and 9 hpf showing that *eve1* expression was gradually downregulated. (**B**) Different concentrations of DMH1 were applied to embryos from 5.3 hpf, which inhibited *eve1* expression. n, total number of embryos analyzed from three independent experiments. (**C**,**D**) qRT-PCR confirmed the expression levels of *eve1* of in situ hybridization in (**A**,**B**). The percentages in each panel in (**A**,**B**) indicate the proportion of embryos displaying the same phenotype as that shown in the photographs of the total embryos examined. Quantitative data are presented as mean ± standard deviation (SD); ****, *p* < 0.0001.

**Figure 3 ijms-24-01733-f003:**
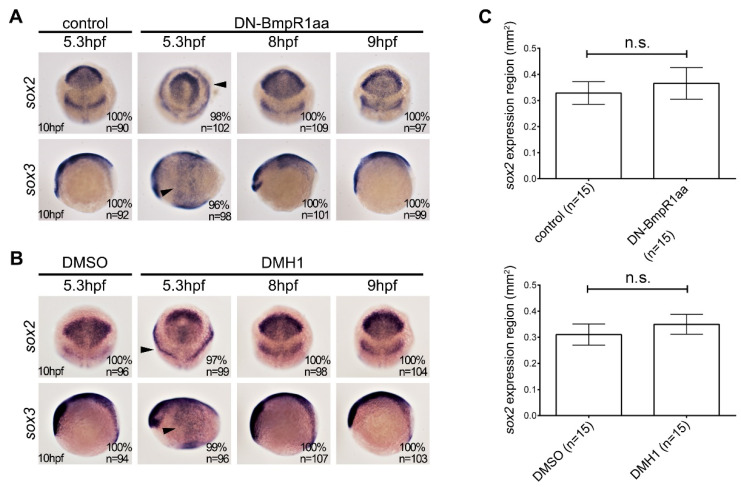
Suppression of BMP signaling after 8 hpf does not affect neural induction. In situ hybridization of heat-shock induced *Tg* (*hsp70l:dnBmpr1aa-GFP*) (**A**) and DMH1-treated (**B**) embryos analyzed at 10 hpf. The upper panels in (**A**,**B**) indicate anterior part of embryos with dorsal to the bottom, and the lower panels indicate the lateral view with anterior to the left. (**A**) Inhibition of BMP signaling by heatshock-induced DN-Bmpr1aa at 5.3 hpf for one hour disrupted the expression pattern and caused ectopic expression of *sox2* and *sox3* (arrowheads), whereas heat shock treatment after 8 hpf did not alter the expression pattern of *sox2* and *sox3*. (**B**) DMH1 treatment at 5.3 for one hour phenocopies the effect of heat-shocked *Tg* (*hsp70l:dnBmpr1aa-GFP*), which is the alternating pattern and ectopic expression of *sox2* and *sox3*. DMH1 treatment after 8 hpf had no effect on the expression pattern of *sox2* and *sox3*. n, total number of embryos analyzed from three independent experiments. (**C**) *sox2*-expressing areas were measured using ImageJ. It should be noted that inhibiting BMP signaling using DN-Bmpr1aa or DMH1 slightly increased *sox2*-expressing areas; however, this was not statistically significant. *n*, total number of embryos analyzed from three independent experiments. The percentages in each panel in (**A**,**B**) indicate the proportion of embryos displaying the same phenotype as that shown in the photographs of the total embryos examined. Quantitative data are presented as mean ± standard deviation (SD). n.s., not significant.

**Figure 4 ijms-24-01733-f004:**
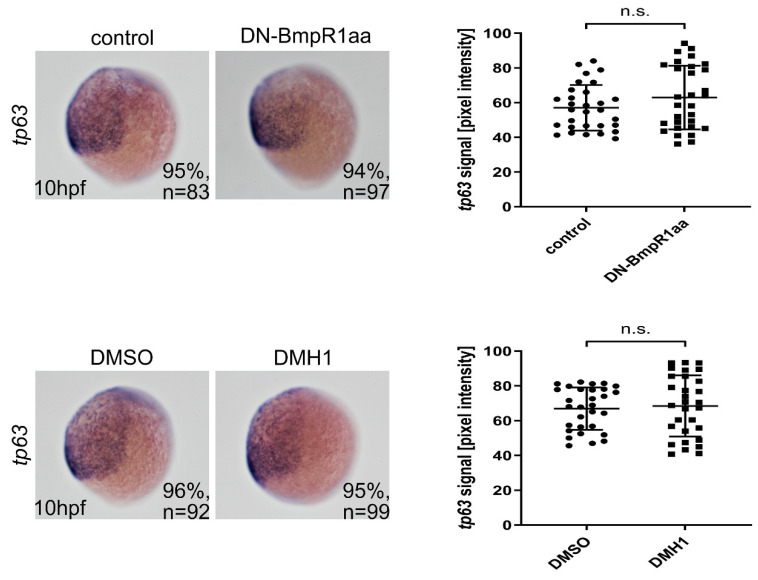
Epidermal ectoderm was unaffected after blocking BMP signaling from 8 hpf. Left panels are lateral views of embryos with ventral to the left. In situ hybridization showed no apparent change in the *tp63*-expressing region in BMP signal-depleted embryos from 8 hpf onward. *n*, total number of embryos analyzed from three independent experiments. The *tp63* signals were measured using ImageJ and quantified (right panels). n, total number of embryos analyzed from three independent experiments. The percentages in each panel indicate the proportion of embryos displaying the same phenotype as that shown in the photographs of the total embryos examined. Quantitative data are presented as mean ± standard deviation (SD). n.s., not significant.

**Figure 5 ijms-24-01733-f005:**
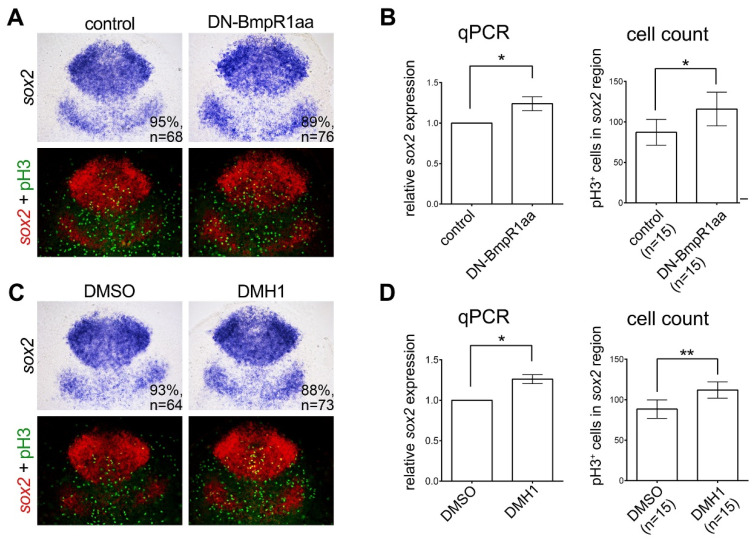
Inhibiting BMP signaling induces the proliferation of neural progenitors. (**A**,**C**) Dorsal view of flat-mounted embryos with the anterior on the top. In situ hybridization of 10 hpf embryos demonstrated increased expression of *sox2* by heat shock-induced DN-Bmpr1aa (**A**) or DMH1 treatment (**C**) at 8 to 10 hpf. The upper panels in (**A**,**C**) are bright-field images and bottom panels are fluorescent images of the same; the expression of *sox2* was pseudo-colored with fluorescent red and counterstained with proliferating marker phospho-histone H3 (pH3) antibody (fluorescent green) to give a better representation of the double-stained cells (fluorescent yellow). (**B**,**D**) The expression of *sox2* was quantified by qRT-PCR (left panel of (**B**,**D**)). pH3-positive cells in the *sox2*-positive areas were manually counted and analyzed, shown on the right panel of (**B**,**D**). n, total number of embryos analyzed from three independent experiments. The percentages in each panel in (**A**,**C**) indicate the proportion of embryos displaying the same phenotype as that shown in the photographs of the total embryos examined. Quantitative data are presented as mean ± standard deviation (SD). * *p* < 0.05; ** *p* < 0.01.

**Figure 6 ijms-24-01733-f006:**
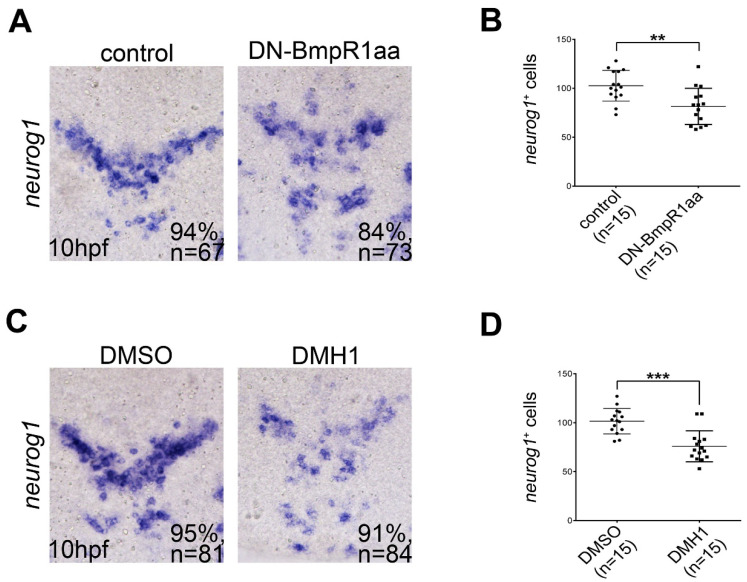
Inhibiting BMP signaling reduces the formation of neuronal precursors. All panels show dorsal views of flat-mounted embryos with the anterior on the top. (**A**,**C**) In situ hybridization of 10 hpf embryos focusing on the midbrain and hindbrain regions showing that *neurog1*-positive cells were reduced by heat-shock-induced DN-BmpR1aa (**A**) or DMH1 treatment (**C**) at 8 to 10 hpf. This was quantified by the cell-count analysis shown in (**B**,**D**), respectively. n, total number of embryos analyzed from three independent experiments. The percentages in each panel in (**A**,**C**) indicate the proportion of embryos displaying the same phenotype as that shown in the photographs of the total embryos examined. Quantitative data are presented as mean ± standard deviation (SD). ** *p* < 0.01; *** *p* < 0.001.

**Figure 7 ijms-24-01733-f007:**
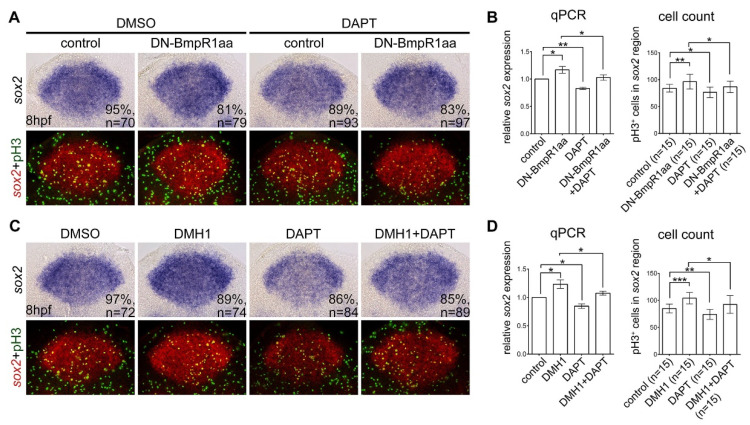
Suppressing Notch signaling abolished the upregulation of neural progenitor proliferation caused by BMP-BmpR1 inactivation (**A**,**C**) All panels are dorsal views of flat-mounted embryos at 10 hpf with the anterior on the top analyzed using in situ hybridization for *sox2* expression and immunohistochemistry with phospho-histone H3 (pH3) antibody. Embryos at 8 hpf were treated with heat-shock-induced DN-BmpR1aa or treated with DHM1 for one hour. The upper panel shows bright field images and the bottom panel shows fluorescent images taken from identical embryos of the upper row; the expression of *sox2* was pseudo-colored with fluorescent red and counterstained with proliferating marker pH3 (fluorescent green). (**B**,**D**) The results of in situ hybridization were confirmed by quantitative reverse transcriptase PCR (qRT-PCR) and the proliferating neural progenitors (fluorescent yellow) were subjected for cell counts (**B**). n, total number of embryos analyzed from three independent experiments. The percentages in each panel in (**A**,**C**) indicate the proportion of embryos displaying the same phenotype as that shown in the photographs of the total embryos examined. Quantitative data are presented as mean ± standard deviation (SD). * *p* < 0.05; ** *p* < 0.01; *** *p* < 0.001.

**Figure 8 ijms-24-01733-f008:**
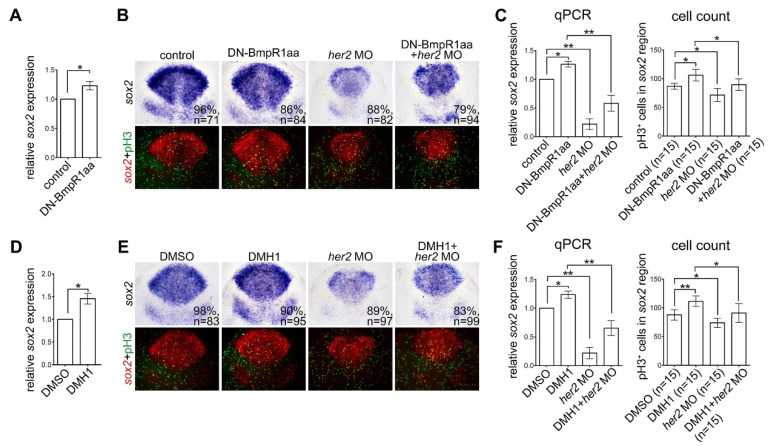
Inactivating BMP signaling upregulates neural progenitor proliferation in response to Her2 knockdown. Heat-shock treatment of Tg (hsp70l:dnBmpr1aa-GFP) or DMH1 was performed at 8–10 hpf embryos and harvested at 10 hpf. (**A**,**D**) qRT-PCR analysis demonstrated increased *her2* expression by activating DN-Bmpr1aa (**A**) and DMH1 treatment (**D**). (**B**,**E**) In situ hybridization of *sox2* expression demonstrating BMP inactivation by activating DN-Bmpr1aa or DMH1 treatment upregulated *sox2* expression and the number of proliferating neural progenitors revealed by *sox2* expression and pH3 counterstaining. Injection of *her2* morpholino at the one-cell stage significantly reduced *sox2* expression and the number of proliferating neural progenitors in embryos with or without BMP inactivation. (**C**,**F**) The results of in situ hybridization were confirmed by qRT-PCR, and the number of pH3-positive cells in *sox2*-expressing areas was counted manually. n, total number of embryos analyzed from three independent experiments. The percentages in each panel in B and E indicate the proportion of embryos displaying the same phenotype as that shown in the photographs of the total embryos examined. Quantitative data are presented as mean ± standard deviation (SD). * *p* < 0.05; ** *p* < 0.01.

## Data Availability

Not applicable.

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
