# Peer review of "Identification of the Time Period during Which BMP Signaling Regulates Proliferation of Neural Progenitor Cells in Zebrafish"

_ijms, 2023, doi:10.3390/ijms24021733_

Round 1

Reviewer 1 Report

Review for: “Identification of the Time Period During Which Bmp Signaling Regulates Proliferation of Neural Progenitor Cells in Zebrafish”

In the present study, the authors discuss the impact of the loss of function of Bmp receptor type 1aa in the neural development of zebrafish. Using a conditional loss-of-function-based approach the authors were able to determine the specific window in which BMP signaling impairs neural development. The impact of BMP signaling in zebrafish neural development has long been described, however in this study the authors were able to add some novel information by detailing a time window in which BMP Signaling affects neural development. However, further experiments should be performed to help the authors defend their purposed hypothesis.

Major comments:

  1. While in this study the authors argue that “we bypassed the effect of Bmp signaling in neural induction”, this is not exclusive to the neural system and also impacts other germ layers and organs, and tissues. Therefore, all similar sentences should be accordingly changed.
  2. The authors suggest that bmpr1aa was presumptively expressed neuroectoderm. However, from in situ analysis alone is difficult to suggest that. Furthermore, other authors have investigated the expression of bmpr1aa and its expression is suggested to be associated with organs and systems such as the heart.
  3. To evaluate the effectiveness of this transgenic line, the authors analyzed the expression of eve1 alone. One single gene expression is not sufficient to make such conclusions and further gene expression analysis should be performed. Moreover, the analysis of BMP direct downstream targets should be analyzed and observed. For instance, a decrease in the expression of SMAD1/5/9 should be also observed
  4. Can the authors provide data that tp63 expression is altered after inhibiting Bmp signaling at earlier time points.
  5. Further information regarding the morpholino should be provided. The authors provide a reference but in that study 2 morpholino was used whereas in this study only 1.
  6. Please provide further information on RNA extraction. The usage of “standard method” is not sufficient and scientifically correct.
  7. The reference list needs to be updated with earlier research manuscripts. Only 3 references are from the last decade with no study after 2017.

Minor comments:

  1. Why did the authors decide to investigate bmpr1ab and acvr1l expression, and what is the relevance to the present study?
  2. “Therefore, to characterize the regulators specifically in the development of neural progenitors, it is essential to temporarily bypass their effect on the earlier developmental event, namely neural induction”. The phrase is not true as there are other options such as transgenic fish.
  3. “Although the function of BMP signaling after neural induction has been intensively studied, most of the aforementioned evidence came from in vitro studies, which could not fully overcome the complex regulation in vivo; thus, the role of BMP signaling after neural induction remains inconclusive.” Also, not true with several other studies documenting such findings in zebrafish
  4. Please describe in the figure legend what the percentage stands for.
  5. RT-PCR stands for reverse transcriptase PCR

Author Response

Reviewer #1

"In the present study, the authors discuss the impact of the loss of function of Bmp receptor type 1aa in the neural development of zebrafish. Using a conditional loss-of-function-based approach the authors were able to determine the specific window in which BMP signaling impairs neural development. The impact of BMP signaling in zebrafish neural development has long been described, however in this study the authors were able to add some novel information by detailing a time window in which BMP Signaling affects neural development. However, further experiments should be performed to help the authors defend their purposed hypothesis."

Major comments:

"1. While in this study the authors argue that “we bypassed the effect of Bmp signaling in neural induction”, this is not exclusive to the neural system and also impacts other germ layers and organs, and tissues. Therefore, all similar sentences should be accordingly changed."

Reply:

Thank you for your comments. We have corrected all the related descriptions. Page 2, paragraph 3, line 1 (“we bypassed the effect of BMP signaling on the regulation of embryonic development before early gastrulation, including dorsoventral patterning, mesoderm formation, and neural induction”); page 5, subheading paragraph 2 (“Inhibiting BMP signaling from 8 hpf bypasses the effects of BMP signaling on dorsoventral patterning and neural induction”); page 5, paragraph 3, line 9 (“the effect of BMP signaling on dorsoventral patterning and neural induction in zebrafish embryos”).

"2. The authors suggest that bmpr1aa was presumptively expressed neuroectoderm. However, from in situ analysis alone is difficult to suggest that. Furthermore, other authors have investigated the expression of bmpr1aa and its expression is suggested to be associated with organs and systems such as the heart."

Reply:

It is indeed true that bmpr1aa is not only expressed in the neuroectoderm but also in early mesoderm and mesodermal derivatives. We have corrected the description accordingly to “was expressed in the presumptive neuroectoderm and developing mesoderm” (page 3, line 1).

"3. To evaluate the effectiveness of this transgenic line, the authors analyzed the expression of eve1 alone. One single gene expression is not sufficient to make such conclusions and further gene expression analysis should be performed."

Reply:

Id3 and Msx1b (previously named Msxb) are direct downstream targets of the BMP–Smad signaling pathway (Hao et al., 2014, Xia et al., 2020, Esteves et al., 2014). We have examined the effect of BMP deficiency on the expression of these genes and demonstrated that heat-shock activation of DN-Bmpr1aa or DMH1 at 5.3 hpf inhibited the expression of id3 and msxb at 6, 7, and 9 hpf, confirming that DN-Bmpr1aa and DMH1 were sufficient to inhibit BMP–Smad signaling. We have included this result in Supplementary Figure S1 and a new paragraph (page 4, paragraph 2).

"Moreover, the analysis of BMP direct downstream targets should be analyzed and observed. For instance, a decrease in the expression of SMAD1/5/9 should be also observed"

Reply:

BMP receptors activate Smad-dependent signaling via Smad1/5/9 phosphorylation. Therefore, we needed to detect phosphorylated Smad1/5/9 to confirm BMP activation. However, the anti-phosphorylated Smad1/5/9 antibody compatible with zebrafish embryos [Cell Signaling (Cat# 9511)] is no longer available from the company, and we could not find a replacement for detecting phosphorylated Smad1/5/9 in zebrafish. Instead, we used id3 and msx1b as direct reporters for Smad activation, as phosphorylated Smad directly binds to the 5ʹ regulatory elements of id3 and msx1b and induces their expression (Xia et al., 2020, Esteves et al., 2014). This result is shown in Supplementary Figure S1.

"4. Can the authors provide data that tp63 expression is altered after inhibiting Bmp signaling at earlier time points."

Reply:

Inhibiting BMP signaling before early gastrulation induces the expansion of the neuroectoderm and loss of the surface ectoderm ((Mullins et al., 1996), as reviewed by (Little and Mullins, 2006) and (Wilm and Solnica-Krezel, 2003)). We performed heat-shock activation of DN-Bmpr1aa or DMH1 treatment at 4 hpf and examined tp63 expression at 50% epiboly (5.3 hpf). We found that inhibiting BMP signaling at this time point was sufficient to inhibit the formation of epidermal ectoderm, as evidenced by reduced tp63 expression. We have included this result in Supplementary Figure S2 and described it on page 5, paragraph 3.

"5. Further information regarding the morpholino should be provided. The authors provide a reference but in that study 2 morpholino was used whereas in this study only 1."

Reply:

Thank you very much for your comment. We have now included this information in the Materials and Methods section (page 12, paragraph 6).

"6. Please provide further information on RNA extraction. The usage of “standard method” is not sufficient and scientifically correct."

Reply:

We have added a new section, 4.6, RNA isolation and cDNA synthesis, to describe RNA extraction in the Materials and Methods section (page 13, paragraph 2).

  1. The reference list needs to be updated with earlier research manuscripts. Only 3 references are from the last decade with no study after 2017.

Reply:

We have added several new references to replace the outdated references (new reference numbers 2, 11, 12, and 29).

"Minor comments:"

"1. Why did the authors decide to investigate bmpr1ab and acvr1l expression, and what is the relevance to the present study?""

Reply:

We initially reviewed the literature to describe the expression of all BMP receptors in zebrafish. The zebrafish genome contains five type I BMP receptors and two type II BMP receptor homologs, namely, bmpr1aa, bmpr1ab, bmpr1ba, bmpr1bb, acvr1l, bmpr2a, and bmpr2b. We found that three of them, bmpr1aa, bmpr1ba, and acvr1l, were expressed in the gastrula and thus performed a detailed expression analysis using in situ hybridization. We discovered that only bmpr1aa was expressed in the presumptive neural ectoderm. Therefore, we agree that the expressions of bmpr1ab and acvr1l are less relevant to the present study and have deleted them from Figure 1 and the text.

"2. “Therefore, to characterize the regulators specifically in the development of neural progenitors, it is essential to temporarily bypass their effect on the earlier developmental event, namely neural induction”. The phrase is not true as there are other options such as transgenic fish."

Reply:

Thank you for your comments. We have deleted the inappropriate description; instead, we have rewritten the description and now focused on the multiple roles of BMP during different embryonic developmental stages to correlate the description with the major theme of the present study (page 1, paragraph 1). We have also discussed the spatial and temporal regulation studies, such as transplantation, tissue-specific promoter-driven gene expression, and electroporation (page 2, line 1).

"3. “Although the function of BMP signaling after neural induction has been intensively studied, most of the aforementioned evidence came from in vitro studies, which could not fully overcome the complex regulation in vivo; thus, the role of BMP signaling after neural induction remains inconclusive.” Also, not true with several other studies documenting such findings in zebrafish"

Reply:

Thank you for pointing this out. It is true that this description is inappropriate and misleading. We have corrected the description and focused on discussing the role of BMP in regulating neural progenitor cells, which is more appropriate for discussing the findings of the present study (page 1, paragraph 1).

"4. Please describe in the figure legend what the percentage stands for."

Reply:

The percentages in each figure indicate the proportion of embryos displaying the same phenotype as that shown in the photographs of the total embryos examined. For example, 80% (n = 90) of 90 embryos were analyzed, and 72 embryos (80%) displayed this phenotype. We have now described this in the figure legends.

"5. RT-PCR stands for reverse transcriptase PCR"

Reply:

Sorry for the typographic error. We have corrected it accordingly.

References

ESTEVES, F. F., SPRINGHORN, A., KAGUE, E., TAYLOR, E., PYROWOLAKIS, G., FISHER, S. & BIER, E. 2014. BMPs regulate msx gene expression in the dorsal neuroectoderm of Drosophila and vertebrates by distinct mechanisms. PLoS Genet, 10, e1004625.

HAO, J., LEE, R., CHANG, A., FAN, J., LABIB, C., PARSA, C., ORLANDO, R., ANDRESEN, B. & HUANG, Y. 2014. DMH1, a small molecule inhibitor of BMP type i receptors, suppresses growth and invasion of lung cancer. PLoS One, 9, e90748.

LITTLE, S. C. & MULLINS, M. C. 2006. Extracellular modulation of BMP activity in patterning the dorsoventral axis. Birth Defects Res C Embryo Today, 78, 224-42.

MULLINS, M. C., HAMMERSCHMIDT, M., KANE, D. A., ODENTHAL, J., BRAND, M., VAN EEDEN, F. J., FURUTANI-SEIKI, M., GRANATO, M., HAFFTER, P., HEISENBERG, C. P., JIANG, Y. J., KELSH, R. N. & NUSSLEIN-VOLHARD, C. 1996. Genes establishing dorsoventral pattern formation in the zebrafish embryo: the ventral specifying genes. Development, 123, 81-93.

WILM, T. P. & SOLNICA-KREZEL, L. 2003. Radar breaks the fog: insights into dorsoventral patterning in zebrafish. Proc Natl Acad Sci U S A, 100, 4363-5.

XIA, L., WANG, S., ZHANG, H., YANG, Y., WEI, J., SHI, Y., ZOU, C., LIU, J., LUO, M., HUANG, A. & WANG, D. 2020. The HBx and HBc of hepatitis B virus can influence Id1 and Id3 by reducing their transcription and stability. Virus Res, 284, 197973.

Reviewer 2 Report

The authors identified that neuronal progenitor cell proliferation is affected by BMP/Notch/Her2 signalling using zebrafish as in vivo model. Transgenic model of zebrafish with BMP modulation was used in this research article to prove the importancy of BMP signaling in neuronal progenitor cell proliferation. The authors found that time window, post-fertilization, is critical for the signaling attenuation.

Minor - the authors sometimes use BMP in all capital letters but sometimes use Bmp. This must be consistent, choose BMP or Bmp.

This article will be of interested for stem cell and neuroscience research communities. The data and results were clearly presented. However, methodology section was not adequately described. The authors shall provide more details in methodology section.

Author Response

"Reviewer #2

The authors identified that neuronal progenitor cell proliferation is affected by BMP/Notch/Her2 signalling using zebrafish as in vivo model. Transgenic model of zebrafish with BMP modulation was used in this research article to prove the importancy of BMP signaling in neuronal progenitor cell proliferation. The authors found that time window, post-fertilization, is critical for the signaling attenuation."

"Minor - the authors sometimes use BMP in all capital letters but sometimes use Bmp. This must be consistent, choose BMP or Bmp."

Reply:

Thank you for your comment. We have corrected this to BMP, all uppercase, throughout the manuscript, to describe BMP signaling in general. For zebrafish homologs, we followed the guideline for Zebrafish Gene Nomenclature, used all letters italicized and lowercase to describe zebrafish genes, and only the first letter was in uppercase and not italicized for zebrafish proteins.

"This article will be of interested for stem cell and neuroscience research communities. The data and results were clearly presented. However, methodology section was not adequately described. The authors shall provide more details in methodology section."

Reply:

We have added description in Sections 4.4. Morpholinos (page 12, paragraph 6) and 4.5. Histological analysis (page 13, paragraph 1, line 6)), and a new paragraph, 4.6 RNA isolation and cDNA synthesis (page 13, paragraph 2), was added in the Materials and Methods section for a detailed description of the methodology.

Reviewer 3 Report

This work certainly deserves high praise, since it contains a clearly formulated hypothesis and a convincingly presented evidence base for the development/confirmation of this hypothesis. The work was performed at a good methodological level using modern molecular biological, immunohistochemical, and genetic methods to assess the involvement of Bmp and Notch signaling pathways in the regulation of proliferation and differentiation of NSC/NCP in zebrafish embryos. When reviewing this work, the following minor questions arose for the authors: 1. As a result of the study of apoptosis of neuronal progenitors with an inactivated signal, data are not shown. However, these data are of considerable interest, and it would be good to present them, if not in the main results, then at least in the supplementary files. 2. It is necessary to give a scheme of the injection of her2 morpholino and control animals, exactly as it was done by the authors in the experiment for possible repetition by other researchers.  

Author Response

Reviewer #3

"This work certainly deserves high praise, since it contains a clearly formulated hypothesis and a convincingly presented evidence base for the development/confirmation of this hypothesis. The work was performed at a good methodological level using modern molecular biological, immunohistochemical, and genetic methods to assess the involvement of Bmp and Notch signaling pathways in the regulation of proliferation and differentiation of NSC/NCP in zebrafish embryos. When reviewing this work, the following minor questions arose for the authors:"

"1. As a result of the study of apoptosis of neuronal progenitors with an inactivated signal, data are not shown. However, these data are of considerable interest, and it would be good to present them, if not in the main results, then at least in the supplementary files. "

Reply:

We have included these results in Supplementary Figure S3 to demonstrate that the apoptosis of neural progenitors was not affected by BMP inactivation.

"2. It is necessary to give a scheme of the injection of her2 morpholino and control animals, exactly as it was done by the authors in the experiment for possible repetition by other researchers."

Reply:

We have rewritten the entire paragraph as section 4.4. Morpholinos (page 12, paragraph 6) for a detailed description of morpholino design and methodology.

Round 2

Reviewer 1 Report

Thank you for addressing most of the comments. Most major issues were addressed.

Minor issues: Part of the text type is written in calibri and another in palatino font type.